# Acute Kidney Failure among Brazilian Agricultural Workers: A Death-Certificate Case-Control Study

**DOI:** 10.3390/ijerph19116519

**Published:** 2022-05-27

**Authors:** Armando Meyer, Aline Souza Espindola Santos, Carmen Ildes Rodrigues Froes Asmus, Volney Magalhaes Camara, Antônio José Leal Costa, Dale P. Sandler, Christine Gibson Parks

**Affiliations:** 1Occupational and Environmental Health Branch, Public Health Institute, Federal University of Rio de Janeiro, Rio de Janeiro 21941-598, Brazil; esp.aline@gmail.com (A.S.E.S.); camaravolney@gmail.com (V.M.C.); 2Maternity-School, School of Medicine, Federal University of Rio de Janeiro, Rio de Janeiro 22240-000, Brazil; carmenfroes@iesc.ufrj.br; 3Epidemiology and Biostatistics Branch, Public Health Institute, Federal University of Rio de Janeiro, Rio de Janeiro 21941-598, Brazil; ajcosta@iesc.ufrj.br; 4Epidemiology Branch, National Institute of Environmental Health Sciences, National Institutes of Health, Research Triangle Park, NC 27709, USA; sandler@niehs.nih.gov (D.P.S.); parks1@niehs.nih.gov (C.G.P.)

**Keywords:** acute renal failure, agricultural occupation, death certificates, kidney diseases, mortality, pesticides

## Abstract

Recent evidence suggests that pesticides may play a role in chronic kidney disease. However, little is known about associations with acute kidney failure (AKF). We investigated trends in AKF and pesticide expenditures and associations with agricultural work in two Brazilian regions with intense use of pesticides, in the south and midwest. Using death certificate data, we investigated trends in AKF mortality (1980–2014). We used joinpoint regression to calculate annual percent changes in AKF mortality rates by urban/rural status and, in rural municipalities, by tertiles of per capita pesticide expenditures. We then compared AKF mortality in farmers and population controls from 2006 to 2014 using logistic regression to estimate odds ratios and 95% confidence intervals adjusted by age, sex, region, education, and race. AKF mortality increased in both regions regardless of urban/rural status; trends were steeper from the mid-1990s to 2000s, and in rural municipalities, they were higher by tertiles of pesticide expenditures. Agricultural workers were more likely to die from AKF than from other causes, especially at younger ages, among females, and in the southern municipalities. We observed increasing AKF mortality in rural areas with greater pesticide expenditures and an association of AKF mortality with agricultural work, especially among younger workers.

## 1. Introduction

Acute kidney failure (AKF; ICD-10: N17) is characterized by an abrupt decrease in renal function that increases toxins and nitrogenous metabolites concentration in the blood [1]. According to the Kidney Disease Improving Global Outcomes organization (KDIGO), AKF is defined by changes in serum creatinine of ≥0.3 mg/dL or ≥26.5 mmol/L within 48 h or increases of ≥1.5 times the baseline within the previous 7 days, or urine volume <0.5 mL/kg/h for 6 h [2]. Biochemical abnormalities in patients with AKF require intensive therapy until metabolic alterations can be reversed, but delayed diagnosis and treatment are some of the factors that make AKF a syndrome with high mortality [3].

A meta-analysis reported geographic differences in the incidence of AKF in hospitalized patients, with the highest rates in southern Europe (31.5%; 95% CI: 23.1–41.3) and South America (29.6%; 95% CI:19.1–42.7), followed by North America 24.5% (95% CI: 21.7–27.5), south Asia 23.7% (95% CI: 7.5–54.4), and eastern Europe 22.0% (95% CI: 9.5–43.3) [4]. Overall, AKF incidence is higher in low-to-middle-income countries than in high-income ones [5,6]. In the United States, racial disparities in AKF incidence rates appear associated with socioeconomic factors [7], and the incidence is higher in the elderly [8,9], consistent with what is seen in other high-income countries where the disease is more common among the elderly. By contrast, low-to-middle-income countries show a higher incidence of AKF among younger adults and children [8,10,11].

Known risk factors for AKF include chronic kidney disease (CKD), nephrotoxic drug ingestion, iodinated contrast, heart failure, liver diseases, sepsis, and diabetes [12,13]. Few environmental or occupational risk factors have been identified. Agricultural pesticides have been associated with CKD [14,15], and growing evidence suggests a positive association of AKF with pesticides and farming activities [16]. Clinical reports have described AKF cases related to accidental or intentional organophosphate (OP) poisonings [17,18]. In a retrospective cohort study, Lee and coworkers [19] observed a higher risk of AKF (hazard ratios (HR) = 6.17, 3.28–11.6 adjusted for age, sex, and comorbidities) among patients with OP poisoning than in controls, especially at younger ages (age 34 years or less HR: 9.65; 95% CI: 4.75–19.6).

Nephrotoxic effects of pesticides have been documented, along with their underlying pathomechanisms, in animal studies. For example, organophosphates insecticides seem to induce changes in epithelial cell and intratubular edema, focal hemorrhage, and inflammatory infiltration in the rat’s kidney (Georgiadis et al. 2018; Kaya et al. 2018). Herbicides’ effects on rat kidneys include marked proximal and distal tubular lesions showing coagulation necrosis with tubular cell loss [20,21,22]. Fungicides have also been related to degeneration of some tubular epithelial cells and hemorrhage in rat kidneys [23,24].

Brazil is one of the top consumers of pesticides in the world. Between 1999 and 2014, pesticide use increased in Brazil from 3.06 to 5.71 Kg/ha (86.6%; average annual increase 4.62%), compared with a global increase from 2.37 to 2.74 Kg/ha (15.6%; average 1.83% annual increase) [25]. Soybean, corn, and sugar cane crops account for about 70% of commercial pesticide use, and the south and midwest regions are the primary users of these substances [26]. Despite the relevance of pesticide use in both areas, the midwest is a region with the highest number of large farms. In contrast, the south region comprises small farms over the total agricultural area [27]. Although there is evidence of acute and chronic adverse health effects of pesticide exposure in Brazilian farmers [28,29,30,31,32], to our knowledge, no studies have examined the association between farming, pesticide exposure, and AKF in Brazil. Our study evaluated time trends in AKF mortality rates in Brazilian south and midwest regions, where most agricultural production is located, for the period 1980–2014. We also compared the AKF mortality risk among southern and midwestern Brazilian agricultural workers with that experienced by the general population in the period 2006–2014.

## 2. Materials and Methods

### 2.1. Studied Area

Agriculture in the midwest is mainly dedicated to the plantation of soybean and other grains on very large farms. Although in the south there are also large farms dedicated to soybean and other grains, in this region there is also a more significant number of medium and small farms, operated by family farmers, which produce many other agricultural commodities. These differences in crop diversity observed in the two regions may result in pesticide use and possible exposure differences in these regions [27,33].

### 2.2. Data Collection

Mortality data were retrieved from the Brazilian Mortality Information System, which uses the International Classification of Diseases’ 10th edition (ICD-10) to organize and codify the main and auxiliary causes of death. Occupations were classified according to the Brazilian Standard Classification of Occupations, which follows the International Classification of Occupations (ISCO). Data on pesticide expenditure on each Brazilian farm in southern and midwestern municipalities were obtained from the Agricultural Census of 1996. National data on pesticide expenditure at the municipality level from the Brazilian agricultural censuses are electronically available for 1985, 1996, 2006, and 2017. We used the 1996 pesticide data to allow time for the exposure to induce biological alterations that could lead to the development and AKF deaths between 2006 and 2014. Mortality data used in the current study are publicly available through the Brazilian Public Health System’s Informatics Department (http://datasus.saude.gov.br/; accessed on 6 April 2022), which does not allow the identification of the cases and therefore does not require Institutional Review Board-IRB review.

### 2.3. Trends in AKF Mortality Sample

We evaluated deaths due to AKF (ICD-10: N17) occurring between 1980 and 2014 in individuals of both sexes aged 20 years or older who lived in a Brazilian south and midwest municipality. Mortality rates were calculated using the number of deaths from AKF in each year of the study period divided by the population of the same year and multiplied by 100,000.

Per capita use of pesticides was calculated by dividing the total pesticide expenditure, in the Brazilian currency of 1996, in each municipality of the south and midwest regions, by its population in the same year.

### 2.4. Death-Certificate-Based Case-Control Sample

We also compared the odds of being an agricultural worker among those who died from AKF against the odds of being an agricultural worker among those who died from other causes. To do so, we retrieved data on all deaths due to AKF (ICD-10: N17) between 2006 and 2014 in individuals 20 years of age or older, of both sexes, who lived in a Brazilian southern and midwestern municipality. Cases (*n* = 6041) were individuals whose primary (underlying) cause of death was AKF, and controls (*n* = 2,010,829) were those who died from any other disease but AKF.

### 2.5. Statistical Analyses

#### 2.5.1. AKF Mortality Time Trend Analysis

AKF mortality rates were calculated for each Brazilian southern and midwestern municipality. They were then grouped into urban and rural status according to the Brazilian Institute of Geography and Statistics’ classification, and for rural states, grouped into low, medium, and high use of pesticides based on the tertiles (T1, T2, and T3) of per capita pesticide expenditure. For each of these groups, we calculated annual AKF mortality rates from 1980 to 2014. We used a joinpoint regression model (Joinpoint version 4.5.0.1; National Cancer Institute, Bethesda, MD, USA; 2017) to identify the years of significant inflection in AKF mortality rates’ linear trends. We also estimated the annual percentage change (APC) and the average annual percent change (AAPC) in AKF mortality rates over the last 15 years of the studied period, stratified by urban/rural status and, within rural municipalities, by tertiles of pesticide use, using Poisson regression that allows for adjusting a data series from the smallest possible number of inflection points [34]. We calculated the AAPC between 2000 and 2014 because mortality data quality in Brazil has been considered better since then [35].

#### 2.5.2. Case-Control Analysis

Logistic regression models were used to calculate odds ratios (OR) and 95% confidence intervals (95% CI) for AKF mortality risk associated with agricultural work, stratified, and adjusted (all other variables) for sex, age (20–50, 51–70, ≥71 years), the region of residence (south and midwest), ethnicity (non-white and white), and education (more than high school and high school or less). Stratified analyses were performed to explore the magnitude of the association between AKF mortality risk and agricultural work within the covariate categories for descriptive purposes but not analytical comparisons. The association of AKF mortality and agricultural occupation was also assessed across 10-year birth cohorts (Appendix A) and 10-year age strata.

## 3. Results

### 3.1. Mortality Trends

In the south, the average annual percent change (AAPC) in AKF mortality for 2000–2014 was 3.41 (95% CI: 1.70, 5.15) (Figure 1a), which was significant in the rural (AAPC: 3.72; 95% CI: 2.14, 5.33) but not in the urban (AAPC: 3.05; 95% CI: 0.82, 5.33) municipalities. Using joinpoint software, we also calculated annual percent changes (APC) for specific periods. Overall AKF mortality rates showed a significant decrease between 1980 and 2006 (APC: −0.88; 95% CI: −1.48, −0.27) but an increasing trend between 2006 and 2014 (APC: 6.75; 95% CI: 3.59, 10.01). In rural municipalities, we observed only a small, non-significant decreasing trend in AKF mortality rates (APC: −0.42; 95% CI: −1.13, 0.30) from 1980 to 2006 compared to a steeper decline for urban areas (−1.82; 95% CI: −2.59, −1.04) for the same period. Subsequently, the parallel rise in both regions maintained higher mortality from AKI in rural compared to urban areas between 2006 and 2014.

In the midwest, the overall AAPC in AKF mortality rates was 4.08 (95% CI: 2.61, 5.58) between 2000 and 2014 (Figure 1b), with a significant increase in both urban and rural municipalities, slightly higher in the latter (Urban: AAPC: 3.57; 95% CI: 1.55, 5.62 and Rural: AAPC: 4.44; 95% CI: 2.22, 6.71). AKF mortality significantly decreased between 1984 and 1993 (APC: −4.53; 95% CI: −7.69, −1.26) and increased between 2006 and 2014 (APC: 5.92; 95% CI: 3.52, 8.37) (Figure 1b). In rural municipalities of midwest Brazil, a significant decrease in AKF mortality (from 1.4 to 0.8 deaths per 100,000 population) between 1984 and 1993 (APC: −4.94; 95% CI: −8.60, −1.14) was followed by two periods of a significant increase in AKF mortality between 1993 and 2006 (APC: 3.05; 95% CI: 1.08, 5.07) and between 2006 and 2014 with APC of 6.32 (95% CI: 1.54, 11.32). Rates were somewhat higher in urban municipalities, but changes in 1984 to 1993 and 1993 to 2006 were less pronounced than in the rural areas.

We then looked at AKF mortality rate trends, across tertiles of per capita pesticide expenditures. In the south (Figure 2a), AKF mortality rates were stable for the first 20 years; joinpoint regression detected an earlier, significant shift in the highest (third) tertile starting in 2000–2014 (APC 5.43; 95% CI: 2.19, 8.79) compared to the second (in 2004) and first (in 2005) tertiles. In the subsequent 9–14 years of follow-up, we observed a significant increase in AKF mortality rates in all three groups. The 2000–2014 AAPC reinforces the more recent increasing trends of AKF mortality rates in the southern municipalities, especially in the higher two tertiles of per capita pesticide expenditure (AAPC 4.56 and 4.13, respectively). In the midwest, AKF mortality rates by pesticide expenditures were more variable (Figure 2b). In the lowest (first) tertile, there was a significant decrease in AKF mortality from 1980 to 1995 (APC: −4.82; 95% CI: −7.93, −1.61), followed by a significant increase (APC:3.98; 95% CI: 2.19, 5.79), while in the third tertile (high), AKF mortality rates showed a slight decrease between 1980 and 2004, followed by a period of significant increase (2004–2014; APC: 5.11; 95% CI: 1.72, 8.61). The average annual percent change (AAPC) for the last 15 years of the studied period showed an increasing trend regardless of per capita pesticide expenditure.

### 3.2. Agricultural Work and AKF

The characteristics of AKF cases and controls are shown in Table 1. Persons who died from AKF (cases) were older and more often from the midwest compared to cases who died from other causes. They were also more likely to be white, lower educated, and agricultural workers.

After adjusting for sex, age at death, region of residence, race/ethnicity, and level of education, persons who died from AKF were more likely to have been agricultural workers than persons who died from other causes (OR (95% CI) 1.23 (1.22–1.42) (Table 2). Agricultural work was associated with AKF mortality in both women (OR: 1.40; 95% CI: 1.23–1.59) and men (OR: 1.27; 95% CI: 1.15–1.39). The odds of being an agricultural worker among those who died from AKF was greatest among younger workers (50 years or less; OR: 1.56; 95% CI: 1.16–2.11) but increased at every age group. The association between agricultural work and AKF mortality was most evident among those from Brazilian southern states (OR: 1.40; 95% CI: 1.29–1.53 versus OR: 1.06; 95% CI: 0.90–1.25 in the midwest states) and appeared stronger among whites (OR: 1.37; 95% CI: 1.26–1.49) than among non-whites (OR: 1.12; 95% CI: 0.94–1.34). The association between agricultural work and death due to AKF did not differ meaningfully by level of education.

Because of the differences observed in the age-stratified analysis, we conducted a further exploratory analysis across the 10-year age strata (Table 3). The association between dying from AKF and agricultural work was highest among the youngest (20–29 years old) agricultural workers (OR: 3.07; 95%: 1.71, 6.26) and lowest among those aged 50–59 years (OR: 0.95; 95% CI: 0.71, 1.26). According to the birth cohort, the odds of being an agricultural worker born in past periods (until 1940–1949) among those who died from AKF were higher than those born after 1950–1959 (Appendix A).

## 4. Discussion

The current study was designed to explore the role of agricultural work and pesticide exposure in the development of acute kidney failure in regions of Brazil with the greatest concentration of agricultural production. In this population-based study in the Brazilian south and midwest, AKF mortality rates showed increasing trends, especially in the last 10–15 years, but regardless of their urban/rural status or level of pesticide expenditure. Although evidence from case reports suggests an association of AKF with pesticides [36,37], few occupational studies have evaluated this relationship. In addition, we observed an association between AKF mortality and agricultural occupation.

CKD is a risk factor for AKF, and the relationship between pesticide exposure and CKD has been evaluated in epidemiological studies. In a prospective analysis of the Agricultural Health Study (AHS) cohort, an increased rate of end-stage renal disease (ESRD) was associated with several specific herbicides in licensed male pesticide applicators [15], while increased risk (HR: 4.22; 95% CI: 1.26, 14.2) was also seen in their female spouses who reported high cumulative overall pesticide use when compared with low pesticide use [14]. CKD of unknown etiology (CKDu) has occurred in Latin America and Sri Lanka, primarily affecting young agricultural workers without known CKD risk factors [38,39,40]. CKDu has been considered multifactorial, with pesticide and metal exposure along with long working hours, intense heat, low fluid intake, and dehydration having been associated with its development [40,41]. A case–control study conducted by Jayasumana et al. [42] found an elevated risk of CKDu (OR 5.12, 95% CI 2.33–11.26) among farmers exposed to the herbicide glyphosate compared with controls. Although there is no evidence of CKDu-induced AKF in rural areas, this hypothesis cannot be ruled out, especially in tropical countries, due to the presence of physical, chemical, and social risk factors. In our study, the youngest agricultural workers (ages 20–29 years) had a higher risk of dying of AKF than from other causes, with a stronger association than in older workers who had higher rates of AKF likely due to other risk factors.

As far as we know, no previous study has assessed trends in AKF mortality in rural areas or areas with pesticide use in Brazil. In our study, rural and urban municipalities and municipalities with low, medium, and high pesticide-per-capita expenditure in the Brazilian south and midwest experienced an increase in AKF mortality in recent years compared to the whole studied period. These results reinforce the hypothesis that different and “traditional” risk factors can contribute to the development of AKF in urban and rural areas. In Brazilian midwestern municipalities, excess mortality from AKF began in the mid-1990s, the same period when there was an increase in pesticide use and planted areas with grains [43]. In the following decades, the midwestern region became the highest pesticide user in terms of liters in Brazil [26], and the progressive and gradual use of these substances was accompanied by a gradual increase in AKF. By contrast, in the south, the increase did not begin until the mid-2000s. The southern region’s long-standing agricultural tradition includes many smallholder farms and diversified crops.

It’s been shown that pesticides have nephrotoxic effects in animal studies. Rats exposed orally to acute and sub-acute levels of the insecticide chlorpyrifos showed tubular dilation, glomerular hypercellularity, and degeneration of renal tubules [44]. Urea nitrogen and creatinine levels were increased in rats treated by oral gavage with the herbicide atrazine [45]. Low oral doses of methyl parathion were related to structural and functional damage to the proximal tubules of male rat kidneys [46]. The herbicide 2,4-D has been related to tubular damage, glomerular alterations, vascular congestion, and an increased number of pyknotic nuclei in the kidneys of rats [21]. In addition, azoxystrobin fungicide administration in rats showed degeneration of tubular epithelial cells and hemorrhage in the intratubular spaces [24].

This study has several limitations. The data on risk factors, such as drug/medication ingestion, iodinated contrast, heart failure, liver disease, infections, and diabetes, were not available. Underreporting and misclassification can introduce biases when comparisons are made of mortality rates among areas or time periods with differences in information quality. In this study, CKD and unspecified kidney failure (UKD; ICD-10: N19) frequency reported as contributory causes of death among AKF cases were 0.54% (33) and 1.27% (77), respectively. Among controls, these numbers were 38,553 (1.92%) for CKD and 39,164 (1.95%) for UKD. However, AKF mortality might include cases of undiagnosed CKD in areas with limited access to healthcare. While the community healthcare system in Brazil has provided basic services to citizens since the mid-1990s, it does not routinely obtain data on pre-clinical laboratory markers of kidney disease. Farmers are exposed to several factors in addition to pesticides that may increase the risk of kidney diseases, and AKF specifically, such as heat and dehydration, metals, infections, and snake bites. In the current study, it was not possible to evaluate which agricultural-related risk factors could be specifically associated with AKF deaths.

A growing number of studies have focused on the acute renal effects of heat stress in agriculture. Moyce and coworkers [47] observed an increased cumulative incidence of AKF (OR: 4.52, 95% CI: 1.61–12.70) after a single day of summer agricultural work, with urine osmolality and creatinine increased among agricultural workers. Another study with the same design found renal alteration, including increased serum creatinine, uric acid, and urea nitrogen, and reduced glomerular filtration rate in sugarcane workers [48]. In rural settings, other common exposures may include leptospirosis, gastroenteritis, and hemolytic-uremic syndrome [5], but these and other individual-level risk factors were not included in our analysis. Overall trends suggest that AKF mortality rates are increasing in rural areas of the southern region, which may similarly be reflected in the stratified regression models showing increased odds of AKF in agricultural workers in this region. In addition to climatic differences (the midwest tends to be dryer) and, therefore, types of crops, specific pesticides and farming practices may also vary. Finally, in rural areas, prior episodes of AKF or pre-existing conditions, such as glomerulonephritis, may go unrecognized, increasing the potential severity of subsequent AKF episodes and associated mortality risk. Complications of pregnancy, such as sepsis and gestational hypertension/preeclampsia, are the other important pre-existing conditions. Excluding deaths prior to age 35 (i.e., most women of reproductive age) did not alter the odds of AKF mortality in females, suggesting pregnancy-related renal failure did not explain the elevated mortality odds in women. Indeed, most AKF mortality in agricultural workers under age 35 occurred in males. The association with agricultural occupation was seen in both males and females, though it appeared somewhat stronger among females. Moreover, we observed an increased chance of AKI mortality in women compared to men in models adjusting for occupation and the other health covariates. Given the possibility of gender-related differences in pathophysiological mechanisms, more research is needed specifically among males and females separately.

The main strength of our study was that we were able to examine AKF mortality related to the farming occupation in a large national database in Brazil, which also allowed us to look at contemporary time trends in AKF mortality across recent decades, a time of increasing pesticide use in Brazil. To evaluate the potential associations of pesticides with AKF mortality, we focused our analyses on the Brazilian south and midwest regions, where pesticide use is very high (mainly used in soybeans, corn, and cotton crops) [26]. Studies that evaluated the risk of renal outcomes in farmers suggest a synergism between environmental factors and increased risk of AKI mortality [49,50]. Our findings reinforce the need for more robust epidemiological studies that account for co-exposures and conditions of agricultural work in the relationship between pesticide exposure and kidney health in Brazil.

## 5. Conclusions

Our results suggest that mortality rates by AKF are increasing in both rural and urban Brazilian municipalities. In the south, AKF mortality rates increased faster in rural municipalities. In addition, in more recent periods (2004–2014), rural municipalities with medium and high pesticide-per-capita expenses showed, in general, higher and faster increases (higher AAPC) in AKF mortality rates. Our study also provides novel and robust evidence of the association between AKF mortality and agricultural work in Brazil, especially among younger workers.

## Figures and Tables

**Figure 1 ijerph-19-06519-f001:**
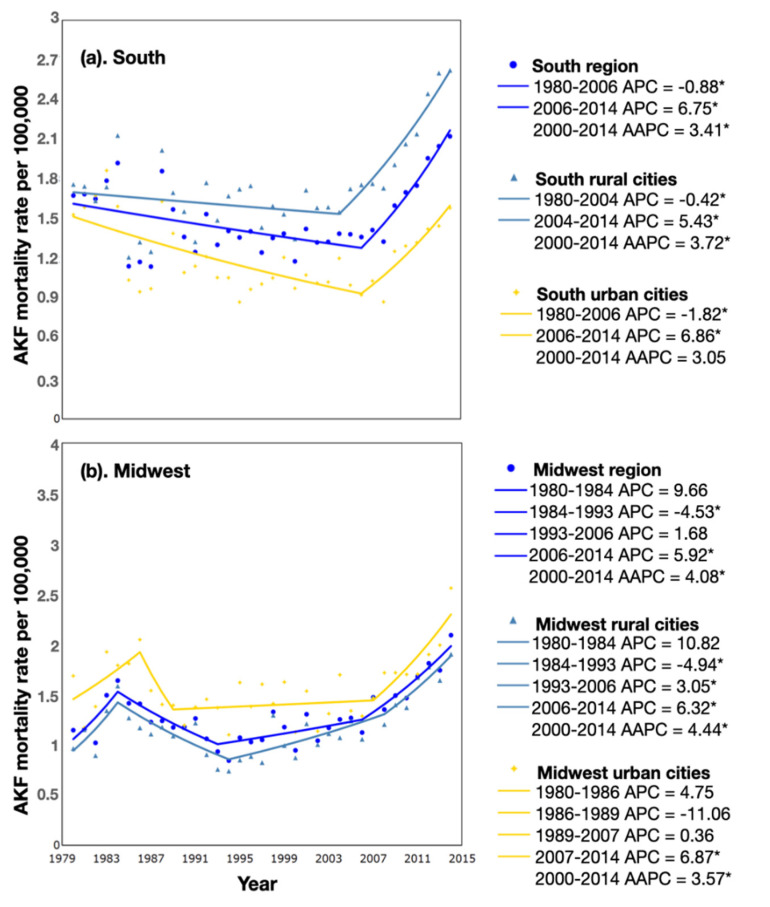
Mortality trends for acute kidney failure, 1980–2014. (**a**) Acute kidney failure mortality crude rates for the south region and its rural and urban municipalities. (**b**) Acute kidney failure mortality crude rates for the midwest region and its rural and urban municipalities. * Significant at *p* < 0.05.

**Figure 2 ijerph-19-06519-f002:**
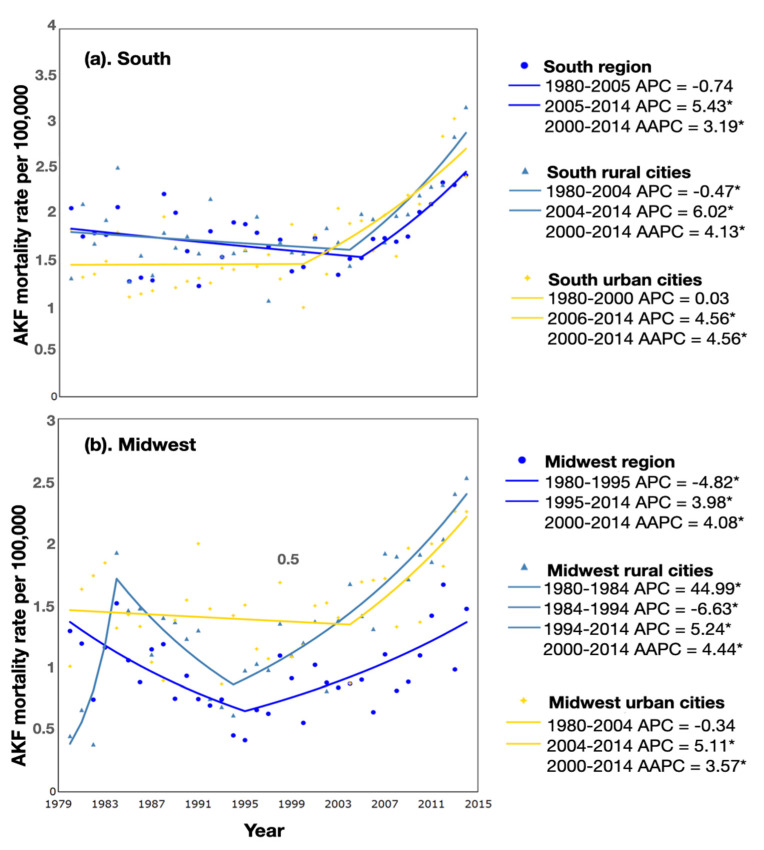
Mortality trends for acute kidney failure by tertiles of pesticide consumption, 1980–2014. (**a**) Acute kidney failure mortality rates for south rural municipalities. (**b**) Acute kidney failure mortality crude rates for midwest rural municipalities. * Significant at *p* < 0.05.

**Table 1 ijerph-19-06519-t001:** Characteristics of AKF cases and controls, based on death certificates for southern and midwestern Brazilian states from 2006 to 2014.

	Cases	Controls	OR * (95% CI)	OR ** (95% CI)
	N (%)	N (%)		
**Sex**				
Male	3215 (53.2)	1,155,237 (57.5)	1.00	1.00
Female	2826 (46.8)	855,142 (42.5)	1.19 (1.13–1.25)	1.05 (0.99–1.11)
**Age at death**				
≤50	539 (8.9)	409,646 (20.4)	1.00	1.00
51–70	1656 (27.4)	660,093 (32.8)	1.91 (1.73–2.10)	1.90 (1.70–2.13)
>70	3846 (63.7)	941,090 (46.8)	3.11 (2.84–3.40)	3.07 (2.76–3.42)
**Region of residence**				
South	4236 (70.1)	1,509,866 (75.1)	1.00	1.00
Midwest	1805 (29.9)	500,963 (24.9)	1.28 (1.22–1.36)	1.37 (1.28–1.47)
**Race/Ethnicity**				
Non-white	1416 (23.4)	502,380 (25.0)	1.00	1.00
White	4625 (76.6)	1,508,449 (75.0)	1.09 (1.03–1.16)	1.21 (1.11–1.30)
**Education**				
More than high school	166 (3.6)	81,147 (5.3)	1.00	1.00
High school or less	4413 (96.4)	1,438,597 (94.7)	1.50 (1.28–1.75)	1.30 (1.11–1.52)
**Occupation *****				
Non-agricultural workers	4030 (79.2)	1,436,019 (83.9)	1.00	1.00
Agricultural workers	1057 (20.8)	274,834 (16.1)	1.37 (1.28–1.47)	1.32 (1.22–1.42)

* Crude odds ratio; ** Odds ratio adjusted by sex, age at death, region of residence, race/ethnicity, education; *** Missing occupation data on 15.8% of cases and 14.9% of controls.

**Table 2 ijerph-19-06519-t002:** Overall, adjusted, and stratified AKF mortality among Brazilian southern and midwestern agricultural workers, 2006–2014.

	Cases	Controls	OR * (95% CI)	OR ** (95% CI)
	N (%)	N (%)		
** Sex **				
**Male**				
Non-agricultural workers	1909 (72.1)	758,357 (78.9)	1.00	1.00
Agricultural workers	739 (27.9)	202,515 (21.1)	1.45 (1.33–1.58)	1.27 (1.15–1.39)
**Female**				
Non-agricultural workers	2121 (87.0)	677,551 (90.4)	1.00	1.00
Agricultural workers	318 (13.0)	72,307 (9.6)	1.41 (1.25–1.58)	1.40 (1.23–1.59)
** Age at death **				
**≤50**				
Non-agricultural workers	369 (86.4)	296,921 (89.2)	1.00	1.00
Agricultural workers	58 (13.6)	35,774 (10.8)	1.31 (0.99–1.72)	1.56 (1.16–2.11)
**51–70**				
Non-agricultural workers	1147(83.4)	474,278 (84.7)	1.00	1.00
Agricultural workers	229 (16.6)	85,743 (15.3)	1.10 (0.96–1.27)	1.13 (0.97–1.32)
**>70**				
Non-agricultural workers	2514 (76.6)	664,820 (81.3)	1.00	1.00
Agricultural workers	770 (23.4)	153,317 (18.7)	1.33 (1.23–1.44)	1.36 (1.25–1.49)
** Region of residence **				
**South**				
Non-agricultural workers	2851 (77.4)	1,098,465 (83.7)	1.00	1.00
Agricultural workers	831 (22.6)	214,252 (16.3)	1.49 (1.38–1.62)	1.40 (1.29–1.53)
**Midwest**				
Non-agricultural workers	1179 (83.9)	337,554 (84.8)	1.00	1.00
Agricultural workers	226 (16.1)	60,582 (15.2)	1.07 (0.93–1.23)	1.06 (0.90–1.25)
** Race/Ethnicity **				
**Non-white**				
Non-agricultural workers	908 (82.7)	342,478 (84.4)	1.00	1.00
Agricultural workers	190 (17.3)	63,539 (15.6)	1.13 (0.96–1.32)	1.12 (0.94–1.34)
**White**				
Non-agricultural workers	3122 (78.3)	1,093,541 (83.8)	1.00	1.00
Agricultural workers	867 (21.7)	211,295 (16.2)	1.44 (1.33–1.55)	1.37 (1.26–1.49)
** Education *** **				
**More than high school**				
Non-agricultural workers	146 (96.7)	71,879 (98.1)	1.00	1.00
Agricultural workers	5 (3.3)	1390 (1.9)	1.77 (0.73–4.33)	1.82 (0.74–4.46)
**High school or less**				
Non-agricultural workers	3036 (76.3)	1,051,486 (81.6)	1.00	1.00
Agricultural workers	941 (23.7)	237,012 (18.4)	1.38 (1.28–1.48)	1.32 (1.22–1.42)

* Crude odds ratio; ** Odds ratio adjusted by sex, age at death, region of residence, race/ethnicity, education; *** Missing data: 31.67% for cases and 32.28% for controls.

**Table 3 ijerph-19-06519-t003:** AKF mortality among Brazilian southern and midwestern agricultural workers, according to 10-year age strata, 2006–2014.

	Cases	Controls	OR * (95% CI)	OR ** (95% CI)
	N (%)	N (%)		
** Age **				
**20–29**				
Non-agricultural workers	50 (83.3)	72,588 (92.3)	1.00	1.00
Agricultural workers	10 (16.7)	6024 (7.7)	2.41 (1.22–4.75)	3.07 (1.71–6.26)
**30–39**				
Non-agricultural workers	92 (82.5)	81,309 (89.7)	1.00	1.00
Agricultural workers	16 (14.8)	9349 (10.3)	1.51 (0.89–2.57)	1.57 (0.84–2.91)
**40–49**				
Non-agricultural workers	196 (87.9)	126,365 (87.6)	1.00	1.00
Agricultural workers	27 (12.1)	17,886 (12.4)	0.97 (0.65–1.46)	1.25 (0.81–1.93)
**50–59**				
Non-agricultural workers	430 (86.7)	196,179 (85.9)		
Agricultural workers	66 (13.3)	32,237 (14.1)	0.93 (0.72–1.21)	0.95 (0.71–1.26)
**60–69**				
Non-agricultural workers	671 (82.4)	264,325 (84.2)	1.00	1.00
Agricultural workers	143 (17.6)	49,687 (15.8)	1.13 (0.95–1.36)	1.16 (0.95–1.41)
**70–79**				
Non-agricultural workers	1049 (76.1)	326,149 (81.7)	1.00	1.00
Agricultural workers	329 (23.9)	72,913 (18.3)	1.40 (1.24–1.59)	1.42 (1.24–1.63)
**80+**				
Non-agricultural workers	1542 (76.8)	369,104 (81.0)	1.00	1.00
Agricultural workers	466 (23.2)	86,738 (19.0)	1.29 (1.16–1.43)	1.31 (1.17–1.47)

* Crude odds ratio; ** Odds ratio adjusted by sex, age at death, region of residence, race/ethnicity, education.

## Data Availability

The data presented in this study are openly available in the DATASUS database at https://datasus.saude.gov.br/transferencia-de-arquivos/, accessed on 6 April 2022.

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
