# Peer review of "Acute Kidney Failure among Brazilian Agricultural Workers: A Death-Certificate Case-Control Study"

_ijerph, 2022, doi:10.3390/ijerph19116519_

Round 1
Reviewer 1 Report
here are my comments about the manuscript:
In the present study, the authors show an interesting association between AKF mortality in Brazilian rural areas and pesticides, presenting engaging data.
They focused their investigation on the Brazilian South and Midwest regions, where pesticide use is very high. Comparing AKF mortality in farmers and population controls in the period 2006-2014, they demonstrated that it was higher in agricultural workers, which more likely died from AKF than from other causes, especially at younger ages.
I only have a minor comment regarding the figures: please upload higher quality figures.
Author Response
REVIEWER #1: I only have a minor comment regarding the figures: please upload higher quality figures.
Response: Thanks for the suggestion. We added higher-quality figures in the manuscript.
Reviewer 2 Report
I recommend the manuscript to be accepted after major revision.
1) In the introduction, there is no literature data on the impact of pesticides on the development of acute kidney failure (AKF). The pathomechanism of AKF development should be described depending on the group of pesticides specified by World Health Organization (WHO). The general data on the development of the AKF are only included in the discussion and in a very limited way.
2) Due to the different pathomechanisms of AKF development in women and men, it is reasonable to provide all data broken down by gender, i.e. age, race/ethinicity, occupation. Extending the analysis would help reduce limitations of the manuscript and clarify some dependencies.
3) There is no information in the introduction as to why the Authors distinguished between the Midwest and the Southern regions of Brasil. Only in the discussion there is one sentence about it (page 9).
4) I do not understand the purpose of providing information on the effects of selected heavy metals on the development of AKF in experimental animals (page 10). In this way, the literature data for individual groups of pesticides should be quoted.
5) I don't understand why the word "traditional" is used in quotation marks (Page 9, sentense: These results reinforce the hypothesis that different and "traditional" risk factors can contribute to the development of AKF in urban and rural area).
6) I propose to replace the ICD-10 with the current one, ie ICD-11.
7) Point 2.1 the mortality data is broken down into two paragraphs, they should be put together.
8) A summary should be prepared, e.g. the last two sentences give the same information.
9) Point 3.1 the first sentences repeat the information contained in the material and methodology (point 2.4).
10) Explain all abbreviations eg HR, IRB, UFD and then use them.
11) Footnotes to tables and figures may not contain abbreviations.
12) Sort key-words in alphabetical order.
13) Standardize the citation system so that it complies with the journal's guidelines.
14) In my version of the manuscript, I did not have access to the supplementary materials.
Author Response
REVIEWER #2:
1) In the introduction, there is no literature data on the impact of pesticides on the development of acute kidney failure (AKF). The pathomechanism of AKF development should be described depending on the group of pesticides specified by the World Health Organization (WHO). The general data on the development of the AKF are only included in the discussion and in a very limited way.
Response: We added in the introduction section the following paragraph on page 2:
Nephrotoxic effects of pesticides have been documented, along with their underlying pathomechanisms, in animal studies. For example, organophosphates insecticides seem to induce changes in epithelial cell and intra-tubular edema, focal hemorrhage, and inflammatory infiltration in the rat's kidney (Georgiadis et al. 2018; Kaya et al. 2018). Herbicides' effects on rat kidneys include marked proximal and distal tubular lesions showing coagulation necrosis with tubular cell loss [20-22]. Fungicides also have been related to degeneration of some tubular epithelial cells and hemorrhage in rat kidneys [23,24].
2) Due to the different pathomechanisms of AKF development in women and men, it is reasonable to provide all data broken down by gender, i.e. age, race/ethnicity, occupation. Extending the analysis would help reduce limitations of the manuscript and clarify some dependencies.
Response: We agreed with the reviewer that there might be gender differences in AKF pathomechanisms. All Tables adjust for gender as well as the other covariates listed, which we have been updated in the table footnotes. Also, in Table 2 we present gender-stratified results for the association between agricultural work and AKF, given the potential differences in occupational exposure for males and females in an agricultural environment. Gender stratification for the overall trends in AKF mortality was not conducted as it extends beyond our primary goal of examining the potential role of pesticides by looking at differences in rural/urban status as well as by pesticide expenditures. The hypothesis proposed by the reviewer that gender stratification could highlight some further results is, however, being explored in a second manuscript. We consider this issue relevant and have added a text on gender differences to the discussion (Page 11).
On page 11 we state:
“The association with agricultural occupation was seen in both males and females, though it appeared somewhat stronger among females. Moreover, we observed an increased chance for AKI mortality in women compared to men in models adjusting for occupation and the other health covariates. Given the possibility of gender-related differences in pathophysiological mechanisms, more research is needed specifically among males and females separately.”
3) There is no information in the introduction as to why the Authors distinguished between the Midwest and the Southern regions of Brasil. Only in the discussion there is one sentence about it (page 9).
Response: We added a sentence in the introduction pointing out the main differences between the Brazilian Midwest and South regions (Page 2 – in red). In addition, to make this information clearer, we also added information about the regions in the methods (page 2).
4) I do not understand the purpose of providing information on the effects of selected heavy metals on the development of AKF in experimental animals (page 10). In this way, the literature data for individual groups of pesticides should be quoted.
Response: We agree with the comment and excluded the paragraph about the effects of selected metals. We have added more information on the nephrotoxicity of specific pesticides in the previous paragraph (Page 10).
5) I don't understand why the word "traditional" is used in quotation marks (Page 9, sentence: These results reinforce the hypothesis that different and "traditional" risk factors can contribute to the development of AKF in urban and rural area).
Response: Marks were excluded.
6) I propose to replace the ICD-10 with the current one, ie ICD-11.
Response: Although it’s an excellent idea, the Brazilian Mortality Information System has not been updated to ICD-11 yet. We hope that we can use the new codes in the near future.
7) Point 2.1 the mortality data is broken down into two paragraphs, they should be put together.
Response: They are together in the revised version.
8) A summary should be prepared, e.g. the last two sentences give the same information.
Response: We could not find the two sentences referred to by the reviewer. Could he/she, please, mention page and lines?
9) Point 3.1 the first sentences repeat the information contained in the material and methodology (point 2.4).
Response: The repeated sentences were deleted (Page 4).
10) Explain all abbreviations eg HR, IRB, UFD and then use them.
Response: We revised all abbreviations used in the manuscript.
11) Footnotes to tables and figures may not contain abbreviations.
Response: abbreviations were removed from tables and figures’ footnotes.
12) Sort keywords in alphabetical order.
Response: Done.
13) Standardize the citation system so that it complies with the journal's guidelines.
Response: We followed the journal’s "Instructions for Authors” (https://www.mdpi.com/journal/ijerph/instructions), where it is written: "References: References must be numbered in order of appearance in the text…” and "In the text, reference numbers should be placed in square brackets [ ], and placed before the punctuation; for example [1], [1–3] or [1,3]. For embedded citations in the text with pagination, use both parentheses and brackets to indicate the reference number and page numbers; for example [5] (p. 10). or [6] (pp. 101–105)."
14) In my version of the manuscript, I did not have access to the supplementary materials.
Response: The supplemental material was uploaded to the journal’s submission system, along with, the rest of the manuscript.
Reviewer 3 Report
This is a well-written article that tests the plausible association between pesticide uses and acute kidney failure induced death. The hypothesis was tested by collecting data from reliable sources and statistical analysis using standard methods. Although this paper did not include comprehensive list of factors that have possible interference, I found it have built up a very good foundation for further studies.
Everything is well explained and discussion is explicitly written. Presentation of data is clear and straightforward. Overall this is a pretty interesting paper, which has its novelty and proposed a possible correlation of environmental substances and human disease. It should have significance in affecting public health.
There is no issue identified.
Author Response
REVIEWER #3: This is a well-written article that tests the plausible association between pesticide uses and acute kidney failure induced death. The hypothesis was tested by collecting data from reliable sources and statistical analysis using standard methods. Although this paper did not include comprehensive list of factors that have possible interference, I found it have built up a very good foundation for further studies. Everything is well explained and discussion is explicitly written. Presentation of data is clear and straightforward. Overall this is a pretty interesting paper, which has its novelty and proposed a possible correlation of environmental substances and human disease. It should have significance in affecting public health. There is no issue identified.
Response: We are grateful for your favorable consideration.
Round 2
Reviewer 2 Report
Correct the captions above the table in supplementary materials so that they do not include abbreviations. And also correct the points so that the numbering is correct in Materials and methods.